# Improving Oral Health with Fluoride-Free Calcium-Phosphate-Based Biomimetic Toothpastes: An Update of the Clinical Evidence

**DOI:** 10.3390/biomimetics8040331

**Published:** 2023-07-27

**Authors:** Hardy Limeback, Joachim Enax, Frederic Meyer

**Affiliations:** 1Faculty of Dentistry, University of Toronto, Toronto, ON M5G 1G6, Canada; 2Dr. Kurt Wolff GmbH & Co. KG, Research Department, Johanneswerkstr. 34-36, 33611 Bielefeld, Germany; joachim.enax@drwolffgroup.com (J.E.); frederic.meyer@drwolffgroup.com (F.M.)

**Keywords:** fluoride-free, toothpaste, calcium phosphate, hydroxyapatite (HAP), casein phosphoprotein-amorphous calcium phosphate (CPP-ACP), beta-tricalcium phosphate (β-TCP), calcium sodium phosphosilicate (CSPS; Bioglass), caries, periodontitis, gingivitis, erosion, whitening

## Abstract

As the demand for clinically effective fluoride-free oral care products for consumers increases, it is important to document which types of toothpastes have been shown in clinical studies to be effective in improving oral health. In this review, we included different indications, i.e., caries prevention, improving periodontal health, reducing dentin hypersensitivity, protecting against dental erosion, and safely improving tooth whitening in defining what constitutes improvement in oral health. While there are several professional and consumer fluoride-containing formulations fortified with calcium-phosphate-based ingredients, this review focuses on fluoride-free toothpastes containing biomimetic calcium-phosphate-based molecules as the primary active ingredients. Several databases were searched, and only clinical trials in human subjects were included; in vitro and animal studies were excluded. There were 62 oral health clinical trials on biomimetic hydroxyapatite (HAP), 57 on casein phosphopeptide-amorphous calcium phosphate (CPP-ACP), 26 on calcium sodium phosphosilicate (CSPS, or so called Bioglass), and 2 on β-tricalcium phosphate (β-TCP). HAP formulations were tested the most in clinical trials for benefits in preventing caries, dentin hypersensitivity, improving periodontal health, and tooth whitening. Based on the current clinical evidence to date, fluoride-free HAP toothpaste formulations are the most versatile of the calcium phosphate active ingredients in toothpastes for improving oral health.

## 1. Introduction

Even in the 21st century, poor oral health remains a major human affliction burdening health care systems in countries all over the world. Dental decay (caries) is still the most common affliction of children and very common in adults [1]. Periodontal disease today is the main reason for tooth loss throughout industrialized countries [2]. However, these human afflictions are preventable with improved diets, healthy nutrition, and especially with improved oral hygiene using toothpastes with active ingredients designed to prevent these common health issues [3]. Furthermore, as teeth are expected to last for a lifetime in ageing populations, dental tissues need to be protected from dental erosion. Some oral care products help protect teeth from mineral loss improving the longevity of the dentition [4]. In addition, people today want whiter and healthier looking teeth. Adults value the cosmetic appearance of their teeth; a whiter dentition improves confidence, improves social acceptance and even employment prospects [5]. Therefore, there is a need to develop active ingredients for toothpastes designed to help with one or more of the preventive roles in home oral care.

Fluoride has been the active ingredient most used in toothpastes throughout the world for the prevention of dental caries for a long time. That fluoride toothpaste reduces dental decay has been documented with many placebo-controlled clinical trials [6]. In order to improve fluoride toothpaste formulations to also help prevent gingivitis and lower the risk of periodontal disease, additional ingredients are added. These include pyrophosphates to help reduce calculus formation [7], bicarbonate for dental plaque removal [8], as well as antibacterial agents such as stannous salts [9], zinc salts [10], and chlorhexidine at low concentrations [11]. Natural ingredients such as herbs and plant-based antimicrobials have also been tested mostly in non-fluoride toothpastes [12].

Fluoridated toothpastes pose safety issues for children under age 6 since there is risk of dental fluorosis from fluoride ingestion [13]. Children under age 3 swallow a significant amount of toothpaste even if they are able to rinse and spit [14]. Because of the risk of fluoride ingestion, dentists in the US and Canada are advised to recommend families with children under the age of 3 year to use a pea-sized amount of fluoridated toothpaste [15,16]. In Europe, children up to age 2 should use a rice-size smear, and those aged 2 to 6 years, a pea size amount [17]. However, children, but also their parents when applying toothpaste for their children, still tend to use more toothpaste, and the majority of those ages ≤ 3 years use it 2 times a day or more often [18]. There is no direct evidence that these smaller amounts of toothpaste can prevent cavities [19]. One study showed that the pea-size amount is less effective in cleaning teeth compared to larger toothpaste amounts [20]. Recent concerns about fluoride’s potential neurotoxicity on developing brains [21,22] have also spurred on research to find alternatives to fluoride as an active ingredient in toothpastes. There is now a concerted effort to find effective non-fluoride anti-caries agents. However, because there is also the need to improve general oral health by also reducing the risk of gingivitis, reducing dentin sensitivity, preventing dental mineral loss, and improving on the appearance of teeth, the active ingredient needs to be very versatile and provide more than one benefit. One ingredient, hydroxyapatite (HAP), has been tested clinically as a general multifunctional useful active ingredient [23].

The most promising candidate active ingredients in toothpastes for achieving all these goals in the future are the calcium-phosphate-based molecules [24]. There is a wide range of these inorganic molecules and the most researched ingredients in this class that have already been tested in toothpastes are amorphous calcium derivatives (casein phosphoprotein-amorphous calcium phosphate, or CPP-ACP), hydroxyapatite, calcium sodium phosphosilicate (CSPS, Novamin, Bioglass), and beta-tricalcium phosphate (β-TCP). A recent review on randomized clinical trials comparing calcium-phosphate-based ingredients was published [25], but the authors omitted clinical evidence from in situ trials, where active ingredients are applied to human enamel slabs imbedded in appliances worn by volunteer subjects. Additionally, the authors did not examine the clinical evidence for hydroxyapatite’s usefulness in controlling caries, even though it has been shown to clinically produce calcium phosphate ions required for remineralization and there have been clinical trials published to show reversal of carious lesions [26].

This review was conducted to examine the clinical evidence published for fluoride-free calcium-phosphate-based toothpastes in order to compare them for determining which one might be a versatile, overall effective toothpaste formulation in promoting good overall oral health.

## 2. Materials and Methods 

A PICO framework was used to guide the search. The following question was posed: “Do fluoride-free toothpastes containing calcium-phosphate-based active ingredients help to improve oral health”? The target populations (P) were humans of all ages. The intervention (I) was using one of the following calcium-phosphate-based active ingredients in a human subject clinical trial, including in situ trials using human enamel imbedded in intra-oral appliances worn by human subjects: amorphous calcium derivatives (casein phosphoprotein-amorphous calcium phosphate, or CPP-ACP), hydroxyapatite (HAP), calcium sodium phosphosilicate, (CSPS, Bioglass) and beta-tricalcium phosphate (β-TCP). The controls (C) were untreated teeth or placebo toothpastes, or positive control toothpastes, and the outcome (O) was one of the following: lowered caries or reduction in white spot lesions, reduced dentin hypersensitivity, protection against dental erosion, improvement of gingival or periodontal health, and/or improved appearance of teeth. The literature was searched using the University of Toronto databases PubMed (Medline), Scopus, and Web of Science, as well as Google Scholar, from inception to 1 June 2023. For the active ingredients, the search terms were “hydroxyapatite”, or “nano-hydroxyapatite”; “casein phosphopeptide-amorphous calcium phosphate” or “CPP-ACP”, or “amorphous calcium phosphate” or “ACP”; “calcium sodium phosphosilicate” or “CSPS” or “bioglass” or “novamin”; “beta-tricalcium phosphate” or “β-TCP” or “tricalcium phosphate” or “TCP”. For the vehicle, the search terms were “toothpaste” or “dentifrice”. For the experimental conditions, the search terms were “in vivo”; ”in situ”; “clinical trial”. For the remineralization outcomes, the search terms were “caries” or “white spot lesion” or “WSL”; “remineralization”; “erosion”. For the dentin hypersensitivity outcomes, they were “sensitivity” or “hypersensitivity”. For the gingival health outcomes, the search terms were “gingivitis” or “gingival” or “periodontal” or “periodontitis”. For the tooth whitening outcomes, the search words were “whiten(s)” or “whitening”.

Inclusion and exclusion criteria: The selection of studies was based on the need to focus on only clinical trials that produced direct clinical evidence for the outcomes directly related to oral health improvement. Animal and in vitro studies were excluded, even those that provide support for the mechanisms of how the active ingredients provide benefits since the evidence needs to be gathered from clinical trials in human subjects. In situ studies were included if the enamel slabs imbedded in appliances worn by volunteer subjects were derived from human (not bovine) enamel. In vivo effects on *Streptococcus mutans* and intra-oral mineral release studies were excluded. All reviews, abstracts, and book chapters were excluded. There were no language restrictions. 

Microsoft Excel spreadsheets of the publications were produced by manually downloading the particulars of each publication of interest (authors, title, journal, abstract, key words) or converting “cvs” files generated by the databases, such as Scopus. The studies were ordered alphabetically, and duplicates were manually removed. Even though the collection of papers was obtained systematically, qualitative syntheses (risk of bias) and quantitative syntheses (meta-analysis) were not carried out. Qualitative (risk of bias) and quantitative (meta-analyses) have been conducted elsewhere on hydroxyapatite-containing oral care products [5,26,27], so the aim of this review was to systematically document the studies published for other fluoride-free calcium phosphate toothpastes, in comparison to the current literature on hydroxyapatite toothpastes, in order to determine the volume and extent of this evidence. Qualitative and quantitative meta-analysis of that literature was not the focus of this review.

## 3. Results

The results of the search are shown in Figure 1. 

A total of 144 clinical trials and in situ clinical studies resulted after applying the exclusion and inclusion criteria. The majority (>80%) of the clinical studies were conducted on HAP- and CPP-ACP-containing toothpastes. Clinical studies on CSPS were mostly on dentin hypersensitivity (DH), and there were only two clinical trials found testing fluoride-free TCP toothpaste. With so many search term combinations, the Google Scholar search yielded an imprecise and excessively large number of titles which, after rapid screening, contained many citations, duplicates, and irrelevant publications. The focus was, therefore, on the titles retrieved in the PubMed, Scopus, and Web of Science databases. Both Scopus and Web of Science permitted “search within results” where subsets of publications were obtained from the large list of publications found using the starting primary search word (e.g., “hydroxyapatite”). 

Appendix A shows the distribution of the clinical studies found using the main databases as a result of the various combinations of search terms. The publications that were retrieved in full and carefully read for each of the calcium-phosphate-based toothpaste active ingredients are summarized in Table 1, Table 2, Table 3 and Table 4. Some studies were cited more than once in the tables because they examined more than one aspect of improving oral health in the same study.

### 3.1. Hydroxyapatite (HAP)

The authors of this review have previously published systematic reviews of the clinical evidence that HAP reduces dental caries [26], reduces dentin hypsersenstivity [27], and improves tooth color [5]. That literature has been updated in this review to include the most recent publications. A total of 62 clinical trials were found where HAP toothpaste was shown to reduce caries, remineralize enamel and protect against erosion, reduce dentin hypesensitivity, improve tooth color, and support gingival health (Table 1).

### 3.2. Casein Phosphopeptide-Amorphous Calcium Phosphate (CPP-ACP)

A total of 57 clinical trials were found on CPP-ACP toothpaste showing that this form of calcium-phosphate-based toothpaste reverses white spot lesions, protects against dental erosion and reduces dentin hypersensitivity (Table 2). Only one study was found where CPP-ACP toothpaste was tested to improve gingival health. Several studies were found to show that CPP-ACP reduced dentin hypersensitivity in studies measuring the effectiveness of professional peroxide bleaching products and that the CPP-ACP did not interfere with the whitening process, but none were found where the active ingredient CPP-ACP was tested on its own in a toothpaste for whitening teeth. 

### 3.3. Calcium Sodium Phosphosilicate (CSPS, Novamin, Biomin, Bioglass)

There have been several studies on fluoride toothpastes fortified with Novamin (CSPS), but those were not summarized in this review since the focus was on fluoride-free toothpastes. Recently, two studies examined CSPS as an active ingredient in fluoride-free toothpastes for controlling caries or white spot lesions [122,145]. There were 23 clinical studies found showing that CSPS was also capable of reducing dentin hypersensitivity. One study was found where CSPS as an isolated active ingredient was able to control gingival health. No studies were found where CSPS toothpastes were tested to improve the color of teeth. These studies are summarized in Table 3.

### 3.4. Beta-Tricalcium Phosphate (β-TCP)

The clinical literature on tricalcium phosphate toothpaste in improving oral health was very limited. While there were a number of in vitro studies and studies conducted on fluoride toothpaste with added TCP (called ‘functionalized’ TCP), only one clinical trial was found where a fluoride-free TCP toothpaste was tested in a clinical trial for reducing caries, and one clinical trial examined how fluorid-free TCP in toothpaste affected dentin hypersensitivity (Table 4).

## 4. Discussion

This systematic review was conducted to compare the clinical evidence that has been published on the calcium-phosphate-containing toothpastes designed to improve oral health. We were interested in comparing the calcium-phosphate-based active ingredients without fluoride. Many fluoride toothpaste formulations contain calcium phosphate additives in an attempt to improve the remineralization and protection of tooth enamel, but recent studies have shown that some ingredients, such as hydroxyapatite, perform as well if not better than fluoridated toothpaste [24,26,27]. Dental fluorosis has been an increasing concern, particularly in those countries that continue to fluoridate their drinking water supplies [169]. In addition, there are concerns that prenatal and even postnatal exposure to fluoride is linked to interference with brain function during early development and growth [170]. For these reasons, it is worthwhile to seek alternatives to fluoridated toothpaste. 

The fluoride-free, calcium-phosphate-containing toothpaste formulations tested in the studies summarized in this review show great promise in that they have been shown in clinical trials to prevent dental decay, reverse white spot lesions, remineralize tooth enamel, protecting it from erosion, desensitize hypersensitive root surfaces and even improve gingival health, all while whitening and brightening the dentition.

There were 62 clinical studies found where HAP was the active ingredient and almost an equal number of clinical studies conducted on CPP-ACP. The vast majority of them used fluoride-toothpaste as positive controls. No study was conducted to compare HAP vs. CPP-ACP in a head-to-head clinical trial. Toothpastes containing CPP-ACP, which contains casein peptides, cannot be used in patients who are allergic to milk proteins. Neither can that toothpaste be given a ‘vegan’ designation. Calcium phosphate ingredients, if accidentally swallowed, are considered safe since they dissociate in the stomach into their constituent inorganic components (calcium and phosphate ions), which are not only harmless but actually contribute to needed dietary sources [171].

One other fluoride-free calcium-phosphate active ingredient that should have been considered but not included in the search was calcium glycerophosphate (CaGP), an active ingredient mentioned in the review by Enax et al. [172] on the remineralization strategies of molar incisor hypocalcification. While this ingredient is used mainly to fortify fluoride toothpaste, it has only been tested in three clinical trials as an active ingredient without fluoride [173,174,175]. In those recent trials, it has been shown to be effective on its own and should really be counted as the fifth active ingredient for fluoride-free calcium-phosphate-containing toothpaste with the potential to reverse white spot lesions. 

## 5. Future Directions

While the clinical evidence to date on the effectiveness of biomimetic fluoride-free calcium-phosphate ingredients in oral care products is already quite extensive and based on dozens of clinical trials, the development of new strategies and products for the prevention and control of oral diseases and maintaining good oral health should continue. Randomized clinical trials (RCTs) where calcium-phosphate-based toothpaste formulations are tested in head-to-head experiments have not been conducted. These would be useful in order to determine which active ingredients most meet the needs of the average consumer in improving overall oral health. Additional clinical trials are required using subjects in susceptible populations and in all age groups. 

## 6. Conclusions

Because of the concern by families of the lasting negative effects of fluoride ingestion with the use of fluoridated toothpaste, there is increased interest by researchers in preventive dentistry to clinically test fluoride-free toothpastes for the potential to be effective in improving oral health. While there is extensive clinical evidence that the biomimetic approach of using hydroxyapatite, casein phopshopeptide-amorphous calcium phosphate, or calcium sodium phosphosilicate has proven successful, additional clinical studies would help identify the most effective active ingredients so that dentists can tailor targeted preventive regimens best suited for patients’ needs. Based on the current clinical evidence to date, fluoride-free hydroxyapatite seems to be an all-round, versatile, and effective agent for improving oral health, in comparison to the other calcium phosphate active ingredients in toothpastes tested clinically.

## Figures and Tables

**Figure 1 biomimetics-08-00331-f001:**
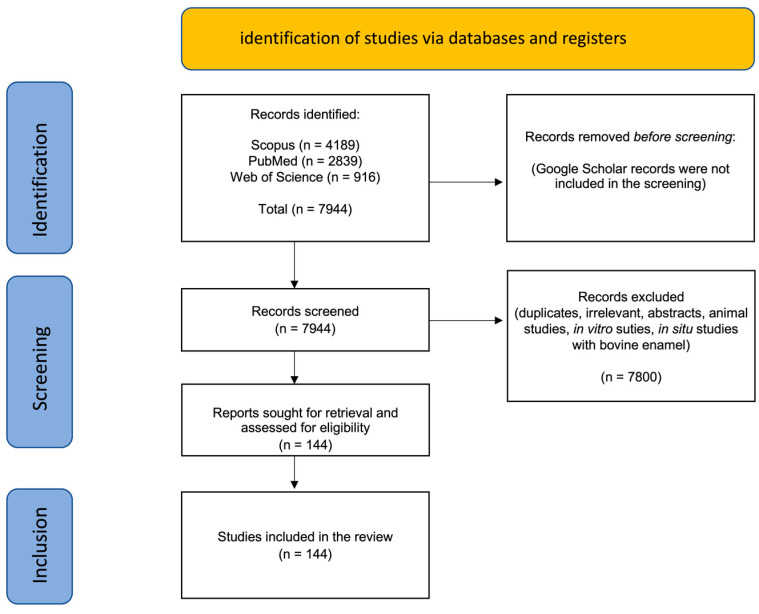
Summary of the search results showing numbers of publications from each database identified and the total records included after screening and exclusion of records.

**Table 1 biomimetics-08-00331-t001:** (**a**) Hydroxyapatite (HAP) clinical trials on reducing caries or white spot lesions (WSL) or preventing erosion, listed chronologically. (**b**) HAP studies in situ using human enamel to measure remineralization or erosion resistance. (**c**) HAP clinical trials on reducing dentin hypersensitivity listed chronologically. (**d**) Hydroxyapatite (HAP) clinical trials on improvement of gingival health listed chronologically. (**e**) Hydroxyapatite (HAP) clinical trials on improving tooth appearance listed chronologically.

(**a**)						
**Study First Author**	**Year**	**Test** **(HAP Used)**	**Controls**	**Trial Subjects,** **Duration**	**Outcome**	**Does HAP Reduce Dental Caries?**
Kani [28]	1989	Apato 5% HAP	Kirara (HAP and F-free)	181 children, -3 years	Significant reduction in DMFT in the HAP group	yes
Lelli [29]	2014	Biorepair (ZnCO_3_/n-HAP)	Sensodyne Pronamel	Extracted premolars after 8 weeks treatment	Zinc-carbonated HAP showed better repair of damaged enamel than fluoride toothpaste	yes
Makeeva [30]	2016	Apadent Total Care	No control	30 subjects -3 mo.	The long-term use of HAP toothpaste increases caries resistance of enamel	yes (but no control)
Schlagenhauf [31]	2019	Karex (10% HAP)	1400 ppm fluoride toothpaste (amine fluoride + SnF_2_)	150 subjects -6 mo.	Works as well as fluoride paste in reducing caries (ICDAS) progression	yes
Bossù [32]	2019	Biorepair (ZnCO_3_/n-HAP)	1. Ordinary toothpaste 2. 500 ppm fluoride toothpaste 3. 1400 ppm toothpaste	40 extracted primary teeth after 15 days trial	Zinc-carbonated HAP showed better remineralization properties than fluoride toothpaste	yes
Badiee [33]	2020	6.7% HAP toothpaste	Fluoride toothpaste	50 post-orthodontic subjects	HAP toothpaste outperformed in fluoride toothpaste in caries reduction	yes
Grocholewicz [34]	2020	ApaCare Repair (10% HAP gel)	(1) ozone (2) not treatment	92 subjects -2 years	HAP gel provided significant reversal of caries	yes
Paszinska [35]	2021	Kinder Karex (10% HAP)	Elmex Kinder Zahnpasta (500 ppm F^−^)	77 children -1 year	HAP and fluoride toothpaste were equivalent in slowing caries progression (ICDAS	yes
Verma [36]	2021	Apagard Premio Toothpaste (10% HAP)	Amflor toothpaste (amine fluoride)	30 orthodontic subjects -15 days	The HAP toothpaste was superior to fluoride toothpaste in restoring the enamel surface post-orthodontic bonding	yes
Butera [37]	2021	MicroRepair (ZnCO_3_/n-HAP)	Sensodyne Repair and Protect	20 subjects with orthodontic buttons -30 days	More deposition of Phosphate and Calcium on the composite resin in the HAP group	likely
Butera [38]	2022	Biorepair Total Protective Repair	Same toothpaste + ZnCO_3_/n-HAP mouthwash	40 rugby players -90 days	Erosion index improved in both test and control	likely
Paszynska [39]	2023	10% HAP	NaF toothpaste (1450 ppm fluoride)	171 adults -18 months	HAP toothpaste was equivalent to fluoride toothpaste in preventing new caries lesions as measured by DMFS and DIAGNOcam	yes
(**b**)						
**Study First Author**	**Year**	**Test** **(HAP Used)**	**Controls**	**Trial Subjects,** **Duration**	**Outcome**	**Does HAP Remineralize Human Enamel?**
Najibfarb [40]	2011	Apagard (5% HAP or 10% HAP)	Crest fluoride toothpaste	30 subjects -28 days per phase	10% hydroxyapatite tooth-paste caused remineralization comparable to a fluoride den-tifrice, inhibiting in situ caries development as effectively as fluoride toothpaste	yes
Amaechi [41]	2019	Karex (10% HAP)	Elmex (Amine fluoride toothpaste, 500 ppm F^−^)	32 subjects wearing appliances with imbedded human enamel blocks -2 mo. (with crossover)	The HAP toothpaste works as well as the fluoride toothpaste in remineralizing enamel	yes
Amaechi [42]	2021	Apagard Deep Care (5% nHAP) along with Apagard M-plus (5% nHAP)	Placebo along with Apagard M-Plus (5% n-HAP)	32 subjects wearing appliances with imbedded human enamel blocks -2 mo. (with crossover)	5% HAP toothpaste remineralized enamel and the added 5% HAP lotion improved the remineralization	yes
Amaechi [43]	2022	Bioniq Repair-Zahncreme (20% HAP)	Colgate Komplett 8 Zahnpaste (1450 ppm F^−^)	15 subjects wearing appliances with imbedded human enamel blocks -1 mo. (with crossover)	HAP toothpaste achieved significantly higher remineralization of MIH lesions than the fluoride toothpaste	yes
(**c**)						
**Study First Author**	**Year**	**Test** **(HAP Product Used)**	**Controls**	**Trial Subjects,** **Duration**	**Outcome**	**Does HAP Toothpaste** **Desensitize Teeth?**
Hüttemann [44]	1987	17% HAP A: with 6 µm particles B: with 2 µm particles	B: 17% salt, C: 0.125% benzocaine, D: placebo, E: 9% HAP. 8% salt, 0.125% benzocaine, F: 17% HAP, 6% SrCl_2_, G: 17% HAP, 5% SrCl_2_, 1% amine fluoride	140 adults -2 weeks	HAP reduced DH over controls	yes
Barone [45]	1991	15% HAP paste	no treatment control	40 adults -24 weeks	reduced DH in the HAP groupbased on before and after measurements	maybe
Park [46]	2005	HAP toothpaste	no treatment control	44 adults -8 weeks	the HAP toothpaste reduced DH	maybe
Kim [47]	2008	Diome Plus PRTC (10% HAP) toothpaste	Strontium chloride toothpaste (Sensodyne GSK)	100 adults -4 weeks	HAP toothpaste worked as well as strontium toothpaste to lower DH	yes
Kang [48]	2009	Diome Plus PRTC (10% HAP) toothpaste	Fluoride toothpaste (2080 Korea) Strontium chloride toothpaste (Sensodyne GSK)	150 adults -4 weeks	HAP toothpaste reduced DH	yes
Kim [49]	2009	Diome Plus PRTC (10% HAP) toothpaste	Strontium chloride toothpaste (Sensodyne GSK)	55 adults -8 weeks	HAP toothpaste worked as well as strontium toothpaste to lower DH	yes
Orsini [50]	2010	Biorepair Plus (30% Zn-carbonate HAP)	Sensodyne Pronamel	75 adults -8 weeks	Zn-Carbonated HAP toothpaste reduced DH	yes
Shetty [51]	2010	A: HAP in dry sol powder B: HAP liquid	C: placebo D: no treatment	45 adults -8 weeks	the HAP toothpaste reduced DH more that the controls	yes
Browning [52]	2012	Renamel nHAP toothpaste	placebo	42 adults -2 weeks	HAP toothpaste reduced DH	yes
Orsini [53]	2013	Biorepair Plus (30% Zn-carbonate HAP)	Colgate Sensitive (8% arginine, MFP at 1450 ppm fluoride) Sensodyne Rapid Relief (8% strontium acetate, NaF at 1044 ppm fluoride)	90 adults -3 days	all three toothpastes reduced DH equally	yes
Jena [54]	2015	NanoXIM (15% HAP)	Vantej (5% Novamin) Sensitive Pro-Relief	45 adults -4 weeks	HAP toothpaste was more effective than 5% Novamin toothpaste	yes
Pinojj [55]	2014	HAP toothpaste (SHY NM)	CSPS toothpaste (SHY) CPP-ACP	80 teeth (adult subjects) -3 months	the HAP and CSPS toothpastes worked better to reduce DH than the CPP-ACP paste	yes
Reddy [56]	2014	Acclaim (15% HAP)	Colgate ProArgin	30 adults 3 days	both toothpastes (HAP and arginine) reduced DH	yes
Vano [57]	2014	Prevdent (15% HAP)	Colgate Cavity Protection (1500 ppm fluoride in MFP) Placebo	105 adults -4 weeks	the HAP toothpaste worked better than the fluoride toothpaste to reduce DH	yes
VJ Narmantha [58]	2014	Acclaim (1% HAP)	Sensodent-K (5% KNO_3_) Propolis	45 adults -4 weeks	HAP toothpaste and Propolis toothpaste both reduced DH	yes
Amin [59]	2015	Acclaim (15% HAP)	none	30 adults -6 mo.	HAP toothpaste reduced DH	maybe
Gopinath [60]	2015	Acclaim (10% nHAP)	Shy-NM (5% CSPS)	36 adults -4 weeks	both HAP and CSPS toothpastes lowered DH	yes
Lee [61]	2015	Denti-guard Sensitive (20% Carbonated HAP, 8% silica)	Sensodyne (10% CaCO_3_, 10% Strontium chloride) Laser treatment	82 adults -4 weeks	HAP toothpaste worked as well as strontium chloride toothpaste and professional laser treatment	yes
Vano [62]	2015	Prevdent (2% HAP in 6% hydrogen peroxide toothpaste)	6% hydrogen peroxide toothpaste control	60 subjects -2 weeks	the HAP added to peroxide toothpaste reduced DH	yes
Makeeva [30]	2016	Apadent Total Care (10% HAP)	No treatment control	30 adults -3 months	HAP toothpaste reduced DH	maybe
Anand [63]	2017	1% nHAP toothpaste	Pro-Argin sensitivity fluoride toothpaste	60 adults -4 weeks	both nHAP and Pro-Argin reduce DH	yes
Makeeva [64]	2018	Innova paste (6% Nano-HAP) + Liquid Enamel (1% Nano-HAP liquid)	No treatment control	40 adults -2 weeks	The combination of HAP toothpaste and HAP mouthwash reduced DH	maybe
Amaechi [65]	2018	Apadent Pro dental cream (20% HAP)	20% silica cream	52 adults -8 weeks	HAP-group showed reduced DH compared to silica	yes
Vano [66]	2018	Cavex Bite & White ExSense (2% nHAP toothpaste)	Colgate Cavity Protection Gel placebo	105 adults -4 weeks	HAP toothpaste reduced DH more than the placebo	yes
Al Asmari [67]	2019	Biorepair (20% Zn-carbonate hydroxyapatite)	no treatment control	72 adults -8 weeks	reduced DH	maybe
Kondyurova [68]	2019	SPLAT Sensitive Ultra (0.5% nHAP)	0.1% nHAP (Splat Professional Sensitive White)	60 adults -4 weeks	both concentrations of HAP reduced DH	yes
Alancar [69]	2020	nHAP toothpaste (± laser)	Placebo + laser Placebo + simulated laser	32 adults -1 mo.	HAP toothpaste reduced DH over control	yes
Ding [70]	2020	Dentiguard Sensitive (20% nanocarbonate-apatite)	placebo	45 adults -6 weeks	HAP toothpaste reduced DH relative to control	yes
Alharith [71]	2021	Nano XIM toothpaste (15% HAP)	Placebo Fluorophat (5% NaF)	63 adults -1 week	HAP toothpaste reduced DH better than fluoride	yes
Amaechi [72]	2021	10% and 15% nHAP toothpaste	10% HAP + 5% KNO_3_ Na-MFP (1400 ppm F-) + CPSC	104 adults -8 weeks	10% HAP ± KNO_3_ reduced DH and 15% HAP worked better than 10% HAP	yes
Ehlers [73]	2021	Kinder Karex (10% HAP)	Elmex Junior (amine fluoride at 1400 ppm fluoride)	21 children -8 weeks	HAP toothpaste worked as well as fluoride toothpaste in lowering DH	yes
Polyakova [74]	2022	20% HAP toothpaste	Zn-magnesium HAP n-FAP toothpaste	30 adults -1 month	the Zn-Magnesium HAP toothpaste worked better that the 20% HAP and n-FAP toothpaste	yes
Vlasova [75]	2022	GARDA SILK (HAP toothpaste with Polyol Germanium Complex)	fluoride toothpaste no toothpaste control	120 adults -2 weeks	HAP toothpaste with PGC reduced DH better than conventional fluoride toothpaste (no HAP-free PGC supplemented placebo used)	maybe
Butera [76]	2022	Biorepair (30% Zn-HAP)	-no treatment control	25 MIH patients -9 mo.	Zn-HAP showed desensitization of MIH teeth	yes
(**d**)						
**Study First Author**	**Year**	**Test** **(HAP Product Used)**	**Controls**	**Trial Subjects,** **Duration**	**Outcome**	**Does HAP Toothpaste** **Improve Gingival Health?**
Harks [77]	2016	Zn-HAP	previously used toothpaste	46 adults -4 weeks	subjective improvement of oral health in both groups (HAP and antibacterial toothpaste)	maybe
Doroshina [78]	2019	Zn-carbonate HAP (CHA)	CSPS toothpaste Herbal toothpaste	25 adults	CHA did not perform as well as the CSPS and herbal toothpastes in reducing gingival health measurements (bleeding on probing)	maybe
Monterubbianesi [79]	2020	Sensitive Ultra Splat	Biorepair Gum Protection Curaprox Enzycal Zero no paste brushing control	80 adults -14 days	all pastes improved gingival health	yes
Steinert [23]	2020	20% Zn-HAP	amine fluoride/stannous fluoride toothpaste	46 subjects -3 months	pocket depth, bleeding on probing improved with both toothpastes	yes
Brauner [80]	2022				the test toothpaste in combination of the mouthwash improved gingival health	not shown directly
Andrea [81]	2022	Biorepair Peribioma	regular toothpaste (no preference)	50 adults -2 months	The HAP toothpaste improved gingival health parameters	yes
(**e**)						
**Study First Author**	**Year**	**Test** **(HAP Product Used)**	**Controls**	**Trial Subjects,** **Duration**	**Outcome**	**Does HAP Toothpaste** **Whiten or Improve the Appearance of Teeth?**
Niwa [82]	2001	3% HAP toothpaste 15% HAP toothpaste	placebo toothpaste	12 adults -1 month	whitening of teeth was achieved (without polishing)	yes
Raoufi [83]	2010	0.1% HAP toothpaste	calcium peroxide toothpaste placebo toothpaste	150 adults -3 months	unable to demonstrate tooth whitening (HAP concentration was too low)	no, concentration was too low
Woo [84]	2014	0.25% HAP toothpaste	0.075% Hydrogen peroxide placebo toothpaste	85 adults -3 months	hydrogen peroxide whitening teeth more than HAP toothpaste	maybe, even at a very low concentration
Bommer [85]	2018	self-assembling peptide matrix and HAP	no treatment control	40 adults -1 month	whitening based on diffuse reflection in vitro was seen in vivo	yes
Steinert [86]	2020	HAP gel		25 adults -1 month	subjective whitening of teeth was achieved by HAP	yes
Steinert [23]	2020	20% Zn-HAP		46 subjects -1 month	patients reported smoother, whiter teeth when using HAP	yes

WSL: white spot lesion; HAP: hydroxyapatite; nHAP: nano-hydroxyapatite; F: fluoride; ICDAS: international caries detection and assessment system; DH: dentin hypersensitivity.

**Table 2 biomimetics-08-00331-t002:** (**a**) Casein phosphopeptide-amorphous calcium phosphate (CPP-ACP) clinical trials on reducing caries or white spot lesions (WSL) or preventing erosion, listed chronologically. (**b**) Casein phosphopeptide-amorphous calcium phosphate (CPP-ACP) studies in situ using human enamel to measure remineralization or erosion resistance, listed chronologically. (**c**) Casein phosphopeptide-amorphous calcium phosphate (CPP-ACP) clinical trials on reducing dentin hypersensitivity listed chronologically. (**d**) Casein phosphopeptide-amorphous calcium phosphate (CPP-ACP) clinical trials on improvement of gingival health. (**e**) Casein phosphopeptide-amorphous calcium phosphate (CPP-ACP) clinical trials on improving tooth appearance.

(**a**)						
**Study First Author**	**Year**	**Test** **(CPP-ACP Used)**	**Controls**	**Trial Subjects,** **Duration**	**Outcome**	**Does F-Free CPP-ACP Clinically** **Reverse WSLs?**
Andersson [87]	2007	Topacal 1st 3 mo.	0.05% NaF rinse + brushing with F toothpaste	26 adolescents -12 mo.	CPP-ACP = 63% complete visual loss of WSL compared to 25% with F	yes
Bailey [88]	2009	CPP-ACP in addition to regular F toothpaste use	Placebo in addition to regular F toothpaste use	45 teens -3 mo.	CPP-ACP = 31% more regression of ICDAS II scores than placebo	yes
Rao [89]	2009	2% CPP	(1) 0.76% Na MFP (2) placebo paste	150 children -24 mo	Both CPP and MFP significantly but equally reduced caries increment compared to placebo	yes
Uysal [90]	2010	Tooth Mousse	(1) Fluoridin N5 (2) placebo	21 orthodontic volunteers donated 60 teeth after 2 mo. trial	Both test groups successfully inhibited caries better than controls	yes
Bröchner [91]	2011	Tooth Mousse in addition to regular F toothpaste use	Pearl Powder gel in addition to regular F toothpaste use	30 subjects, -12 mo.	Reduction in QLF for WSL for both test and control	yes
Akin [92]	2012	Tooth Mousse	(1) brush (2) 0.025% F rinse	80 subjects, -6 mo.	CPP-ACP = 58% reduction in WSL area (1) 45%, (2) 48% (3) micro-abrasion 97%	yes
Sitthisettapong [93]	2012	10% CPP-ACP in addition to regular F toothpaste use	Regular toothbrushing with a F paste	296 preschoolers -12 mo.	ICDAS scores were no difference between test and control	no
Wang [94]	2012	Tooth Mousse	1100 ppm F paste	20 orthodontic patients -6 mo.	CPP-ACP significantly reduced WSL as measured by enamel decalcification index- the F paste control did not	yes
Krithikadatta [95]	2013	10% CPP-ACP	(1) MI Paste plus 0.2% NaF (2) 0.5% NaF mouth rinse	45 subjects -1 mo.	Both CPP-ACP groups had fewer WSL (visual) with decrease in DIAGNOdent readings compared to the control	yes
Plonka [96]	2013	10% CPP-ACP along with 0.304% F paste	0.12% Chlorhexidine (CHX) along with 0.304% F paste	622 children -24 mo.	No significant caries increment benefit over fluoride for CPP-ACP or CHX	no
Aykut- Yetkiner [97]	2014	Tooth Mousse	F toothpaste	60 children -3 mo.	CPP-ACP provided a slight remineralization effect as measured by DIAGNOdent compared to F paste	yes
Yazıcıoğlu [98]	2014	Tooth Mousse	(1) no treatment (2) ozone (3) APF gel (4) Clearfil Protect Bond	125 approximal lesions -18 mo	All groups arrested approximal lesions compared to the non-treatment group	yes
Zhang [99]	2014	CPP-ACP	Fluoride varnish (Duraphat)	112 head cancer patients -12 mo	CPP-ACP reduced radiation caries more than FV	yes
Llena [100]	2015	CPP-ACP	(1) CPP-ACFP (2) fluoride varnish (FV) monthly	786 WSLs in children 3 mo.	CPP-ACP reduced DIAGNOdent and ICDASII scores, but CPP-ACFP and FV were superior	yes
Memarpour [101]	2015	Tooth Mousse	(1) OHI, dietary counseling (2) (1) + fluoride varnish (FV) (3) no treatment	140 children -12 mo.	CPP-ACP reduced the size of WSL and produced smaller increases of dmft scores compared to counselling and FV	yes
Sitthisettapong [102]	2015	10% CPP-ACP in addition to regular F toothpaste use	Regular toothbrushing with a F paste	103 children -12 mo.	There was significant reduction in QFL but no significant difference between test and control	no
Sim [103]	2019	CPP-ACP along with 0.4% SnF2 + 0.32% NaF paste	Placebo	24 head and neck cancer patients -3 mo.	The test subjects had a 51% reduction in caries as measured by ICDASII	yes
Esenlik [104]	2016	Tooth Mousse	No other treatment	57 patients -12 mo.	CPP-ACP significantly reduced WSLs	yes
Güçlü [105]	2016	CPP-ACP	(1) 5% NaF varnish (FV) (2) FV + CPP-ACP (3) no treatment	21 children -3 mo.	Control FV< CPP-ACP or CPP-ACP + FV in laser fluorescent and visual assessment of WSLs	yes
Munjal [106]	2016	Tooth Mousse	No treatment, no orthodontics	679 WSLs (20 treatment group children) -3 mo.	CPP-ACP significantly reduced the WSLs compared to controls according to computerized image analysis	yes
Singh [107]	2016	10% CPP-ACP in addition to regular F toothpaste use	(1) Fluoride varnish in addition to regular toothpaste use (2)regular fluoride toothpaste use	45 subjects post orthodontics -6 mo.	Both FV and CPP-ACP were more effective than F-toothpaste in reducing WSLs (visual, DIAGNOdent readings)	yes
Karabekiroğlu [108]	2017	10% CPP-ACP	F toothpaste	41 subjects -36 mo.	CPP-ACP was not better than regular F paste in reducing WSLs as measured by DIAGNOdent, Gorelik index, ICDAS II	no
Mendes [109]	2018	CPP-ACP	(1) CPP-ACP + fluoride (2) F gel (3) placebo paste	36 children -3 mo.	All treatments produced decreased DIAGNOdent readings, with the best result obtained with CPP-ACP + F	yes
Wang [110]	2018	Tooth Mousse in addition to regular F toothpaste use	(1) F paste + 0.01% F mouth rinse (2) F paste only	21 orthodontic patients -6 mo.	WSL areas were reduced in all groups and the CPP-ACP had the greatest effect	yes
Bobu [111]	2019	10% CPP-ACP	(1) CPP-ACFP (2) 2–10% CPP-ACP + 0.2% NaF paste (3) 0.05% NaF mouth rinse (4) control	80 subjects -3 mo.	All treatment groups significantly lowered DIAGNOdent readings and visual appearance of early caries lesions	yes
Tingyun [112]	2019	MI Paste	(1) F-free placebo (2) OHOLV toothpaste	15 orthodontic patients donated 60 premolars after 10-day treatment	CPP-ACP and OHOLV produced higher calcium and phosphate levels in demineralized enamel	yes
Al-Batayneh [113]	2020	Tooth Mousse	(1) 500 ppm F toothpaste (2) 1 + Tooth Mousse	114 children -6 mo.	CPP-ACP = fluoride paste in reducing QFL, WSL area -CPP-ACP not a booster for F paste)	yes
Bangi [114]	2020	Tooth Mousse	(1) Colgate Strong toothpaste (2) Colgate Phos-Flur mouthwash (3) SHY-NM (CSPS glass paste)	80 subjects, -6 mo.	All significantly reduced WSL decalcification index, but CPP-ACP outperformed the others	yes
Perić [115]	2020	CPP-ACP	(1) CPP-ACFP (2) 0.05% NaF mouth rinse	30 Sjögren’s patients -6 mo	Reduction in WSL in all groups, but no significant difference in DMFS	yes
Ashour [116]	2021	Tooth Mousse in addition to regular F toothpaste use	(1) Tooth Mousse Plus + F toothpaste (2) F toothpaste only	51 subjects -6 mo.	All treatment groups provided slight remineralization as judged by Vistacam scores	yes
Juárez-López [117]	2021	CPP-ACP in addition to regular F toothpaste use	(1) Chewing gum with CPP-ACP (2) F toothpaste only	90 children -3 mo.	CPP-ACP in chewing gum was more effective than CPP-ACP cream in decreasing fluorescence	yes
El-Sherif [118]	2022	CPP-ACP	Pearl powder	57 subjects -3 mo.	CPP-APP and pearl powder both reduced WSL areas and improved their color	yes
Hamdi [119]	2022	CPP-ACP	(1) SDF-KI (2) tricalcium silicate (TCS)	45 patients -24 mo.	Both CPP-ACP and TSC reduced DIAGNOdent readings. SDF-KI significantly remineralized early carious lesions	yes
Olgen [120]	2022	CPP-ACP	(1)CPP-ACFP (2) fluoride varnish (FV)	49 children with MIH -24 mo.	All treatments significantly reduced DIAGNOdent and ICDAS scores with no significant difference between them	yes
Salah [121]	2022	CPP-ACP	(1) BiominF (2) Novamin	60 orthodontic subjects -6 mo.	ICDASII scores, WSL areas and DIAGNOdent scores were reduced by all treatments-BiominF was best	yes
Simon [122]	2022	Tooth Mousse	ICON resin infiltration	60 children -12 mo.	Both treatments reduced WSL areas using ICDASII scores, digitized photos	yes
(**b**)						
**Study First Author**	**Year**	**Test** **(CPP-ACP Product Used)**	**Controls**	**Study Design**	**Outcome**	**Does F-Free CPP-ACP** **Remin-** **eralize Human Enamel?**
Srinivasan [123]	2010	CPP-ACP	(1) CPP-ACFP (2) saliva placebo	5 volunteers wearing human enamel slabs imbedded in appliances	CPP-ACFP remineralized the enamel slabs better than CPP-ACP and both were substantially better than saliva	yes
Shen [124]	2011	Tooth Mousse (TM)	(1) 1000 ppm F paste (2) Clinpro with 950 ppm F (3) 5000 ppm F paste (4) Tooth Mousse + 900 ppm F (TMP) (4) placebo	Volunteers wearing human enamel slabs in appliances	TMP was better than TM and both were better at remineralization than ClinproF or 5000 ppm F paste as measured by transverse microradiography	yes
Perić [125]	2015	CPP-ACP	(1) CPP-ACFP (2) 0.05% NaF mouth rinse	30 Sjögren’s patients -enamel slabs on appliances 1 mo.	Both CPP-ACP agents reduced enamel defects better than NaF mouthrinse	yes
Garry [126]	2017	Tooth Mousse (along with F toothpaste)	F toothpaste control	12 patients wearing fixed orthodontic appliances	CPP-ACP significantly improved remineralization as measured by transverse microradiography	yes
Zawaideh [127]	2017	Tooth Mousse	(1) Pronamel (2) no treatment	20 subjects wearing appliances with human enamel slabs from permanent and primary teeth	CPP-ACP and fluoride protected against dental erosion as measured by surface microhardness	yes
Yu [128]	2018	Tooth Mousse	Water control	12 volunteers wearing human enamel slabs in appliances	CPP-ACP reduced erosion as measured by microhardness	yes
de Oliveira [129]	2020	Mi Paste	(1) MI Paste Plus (2) 1000 ppm fluoride toothpaste (3) placebo toothpaste	10 participants -four 10-day experiments	Remineralizing agents (MP, MPP, and DF) were able to inhibit demineralization of human enamel subjected to high cariogenic challenge in situ.	yes
de Oliveira [130]	2022	MI Paste	(1) MI Paste Plus (2) 1000 ppm fluoride toothpaste (3) placebo toothpaste	10 participants -four 10-day experiments	CPP-ACP and fluoride both prevent demineralization as measured by microhardness	yes
Kumar [131]	2022	CPP-ACP	Fluoride varnish (FV)	30 subjects wearing ortho appliances with human MIH enamel slabs -6 month	CPP-ACP = FV in remineralizing MIH enamel	yes
(**c**)						
**Study First Author**	**Year**	**Test** **(CPP-ACP Product Used)**	**Controls**	**Study Design**	**Outcome**	**Does F-Free CPP-ACP** **reduce tooth sensitivity?**
Borgess [132]	2012	CPP-ACP	No sensitivity treatment	3 patients -some teeth were treated with 20% carbamide peroxide with CPP-ACP	CPP-ACP reduce sensitivity compared to no treatment (pilot study)	yes
Özgül [133]	2013	MI Paste	(1) CPP-ACFP (2) CPP-ACFP + ozone (3) fluoride varnish (Bifluorid) (4) FV + ozone (5) CPP-ACP + ozone	42 MIH patients -3 mo.	Ozone prolonged the desensitization effect of CPP-ACP and FV, but not CPP-ACFP -all 3 effectively reduced tooth sensitivity	yes
Maghaireh [134]	2014	10% CPP-ACP	(1) 2% NaF gel (2) placebo gel	51 patients after bleaching -14 days	CPP-ACP can lower sensitivity post bleaching as well as F	yes
Mahesuti [135]	2014	MI Paste	(1) UltraEZ (KNO3) (2) UltraEZ placebo (3) MI Paste placebo	102 subjects -2 mo.	MI Paste has sustained pain relief compared to KNO3	yes
Zhang [136]	2014	CPP-ACP	Fluoride varnish (Duraphat)	112 head and neck cancer patients -12 mo	CPP-ACP reduced post radiation tooth sensitivity more than FV	yes
Konekeri [137]	2015	CPP-ACP	(2) KNO3 treatment	48 patients -6 weeks	CPP-ACP was better at reducing tooth sensitivity than KNO3	yes
Nanjundasetty [138]	2016	Tooth Mousse	(1) Sensodyne KF (2) placebo	69 fluorosis patients -10 min. after each bleaching session (2) -7 days	MI Paste and Sensodyne equally reduced tooth sensitivity compared to the placebo	yes
Tarique [139]	2017	CPP-ACP	(1) 5% NaF varnish (2) 5% KNO3	36 patients after bleaching -10 day for 3 mo.	CPP-ACP effectively reduced tooth sensitivity more than the other two test groups	yes
Pasini [140]	2018	CPP-ACP	F paste	40 MIH patients -3 mo.	CPP-ACP reduced tooth sensitivity compared to the F paste control	yes
Yassin [141]	2019	CPP-ACP	Placebo paste	24 patients -custom tray application 30 min/day, 7 days after bleaching	CPP-ACP effectively reduced tooth sensitivity compared to the placebo paste	yes
Adil [142]	2021	CPP-ACP	(1) KO3 + Na MFP (2) placebo gel	2011 patients -12 hr. for 3 days after bleaching	CPP-ACP and F effectively reduced tooth sensitivity	yes
Gümüştaş [143]	2022	CPP-ACP	(1) HAP (2) NaF gel	64 subjects -4 min application before bleaching	HAP and F treatments reduced sensitivity, CPP-ACP did not	no
(**d**)						
**Study First Author**	**Year**	**Test** **(CPP-ACP Product Used)**	**Controls**	**Trial Subjects,** **Duration**	**Outcome**	**Does CPP-ACP Toothpaste** **Improve Gingival Health?**
Perić [115]	2020	CPP-ACP toothpaste	CPP-ACPP (with 0.5% NaF) toothpaste 0.5% NaF toothpaste	30 Sjögren’s patients -4 weeks	no significant improvement in gingival health but improvement in dry mouth symptoms	not shown
(**e**)						
No studies were found.						

CPP-ACP: 10% casein phosphoprotein-amorphous calcium phosphate (MI Paste, Tooth Mousse); CPP-ACFP: 10% casein phosphoprotein-amorphous calcium phosphate with added fluoride to 900 ppm (MI Paste Plus, Tooth Mousse Plus); F: fluoride; FV: fluoride varnish; WSL: white spot lesion; QFL: quantitative fluorescent light; ICDAS II: international caries detection and assessment system (modified from ICDAS I).

**Table 3 biomimetics-08-00331-t003:** (**a**) Calcium sodium phosphosilicate (CSPS) clinical trials on reducing caries or white spot lesions (WSL) or preventing erosion, listed chronologically. (**b**) Calcium sodium phosphosilicate (CSPS) in situ clinical trials on reducing caries on preventing erosion. (**c**) Calcium sodium phosphosilicate (CSPS) clinical trials on reducing dentin hypersensitivity listed chronologically. (**d**) Calcium sodium phosphosilicate (CSPS) clinical trials on improvement of gingival health.

(**a**)						
**Study First Author**	**Year**	**Test** **(CSPS Used)**	**Controls**	**Trial Subjects,** **Duration**	**Outcome**	**Does CSPS Toothpaste Reduce Dental Caries?**
Salah [121]	2022	Biomin slurry and toothpaste	BiominF slurry and toothpaste CPP-ACP toothpaste	60 orthodontic patients -4 weeks	All three reduced WSL, with BiominF performing the best	yes
Tiwari [144]	2023	NovaMin toothpaste	probiotic toothpaste fluoride toothpaste	93 orthodontic patients -6 months	All three toothpastes reduced WSLs (the probiotic toothpaste group had the least WSLs)	yes
(**b**)						
No studies were found.						
(**c**)						
**Study First Author**	**Year**	**Test** **(CSPS Used)**	**Controls**	**Trial Subjects,** **Duration**	**Outcome**	**Does CSPS Toothpaste Desensitise Teeth**
Du [145]	2008	NovaMin Toothpaste (2.5% and 7.5% CSPS)	placebo toothpaste Strontium chloride toothpaste	71 adults -6 weeks	NovaMin reduced SDH better than placebo and strontium chloride toothpaste	yes
Litkowski [146]	2010	NovaMin Toothpaste (2.5% and 7.5% CSPS)	placebo toothpaste	66 adults -8 weeks	NovaMin reduced SDH better than placebo	yes
Narongdej [147]	2010	NovaMin powder and toothpaste	placebo powder + NovaMin toothpaste placebo powder + fluoride/KNO_3_ toothpaste	60 adults -4 weeks	NovaMin powder and toothpaste reduced DH better than the Potassium nitrate/fluoride toothpaste	yes
Pradeep [148]	2010	NovaMin toothpaste SHY-NM (5% CSPS)	placebo potassium nitrate toothpaste	110 adults -6 weeks	NovaMin reduced DH better than the placebo and potassium nitrate toothpastes	yes
Salian [149]	2010	NovaMin (5% CSPS)	5% potassium nitrate toothpaste placebo toothpaste	30 adults -4 weeks	NovaMin reduced DH better than the placebo and potassium nitrate toothpastes	yes
Sharma [150]	2010	NovaMin (7.5% CSPS)	5% potassium nitrate toothpaste 0.4% Stannous fluoride toothpaste	120 subjects -12 weeks	All three reduced DH but NovaMin worked better than the others at early time points	yes
West [151]	2011	NovaMin (5% CSPS)	8% arginine toothpaste water control placebo toothpaste	volunteers wore appliances with dentin slices -4 days	NovaMin showed better dentin occlusion and retention than the arginine toothpaste	yes
Pradeep [152]	2012	Novamin SHY (5% CSPS)	5% potassium nitrate toothpaste 3.88% amine fluoride toothpaste placebo toothpaste	149 adults -6 weeks	The Novamin toothpaste showed better results than the others in lowering DH	yes
Rajesh [153]	2012	Novamin SHY (5% CSPS)	Pepsodent toothpaste	30 adults -8 weeks	NovaMin reduced DH better than the placebo toothpaste	yes
Surve [154]	2012	CSPS toothpaste	potassium nitrate toothpaste	20 adults -8 weeks	both reduced DH	yes
Acharya [155]	2013	CSPS toothpaste	potassium nitrate toothpaste	20 adults -8 weeks	both reduced DH but the CSPS toothpaste worked better early in the in the trial	yes
Jena [54]	2015	Vantej (NovaMin 5% CSPS)	Colgate Sensitive Pro-Relief -8% arginine with fluoride) nanoXIM (15% HAP)	45 adults -4 weeks	all three reduced DH, but nHAP toothpaste performed the best	yes
Pintado-Palomino [156]	2015	Bioglass 45S5	7.5% Biosilicate toothpaste Sensodyne toothpaste Odontis RX Sensi Block toothpaste Desesibilize Nano P (HAP toothpaste	140 adults -2 weeks	Toothpaste containing Bioactive glass reduced tooth sensitivity caused by vital bleaching	yes
Samuel [157]	2015	NovaMin toothpaste	ProArgin toothpaste Gluma Desensitizer	147 adults -1 month	ProArgin toothpaste and Gluma sealer reduced DH from a single application compared to NovaMin	yes
Majji [158]	2016	NovaMin (5% CSPS)	5% potassium nitrate toothpaste 10% strontium chloride herbal toothpaste	160 adults -2 months	the CSPS toothpaste showed better reduction in DH than the others	yes
Sufi [159]	2016	5% CSPS	placebo CSPS fluoride toothpaste	134 adults -8 weeks	small and inconsistent outcomes	no
Sufi [160]	2016	5% CSPS	placebo CSPS fluoride toothpaste	134 adults -8 weeks	CSPS paste reduced DH similar to placebo	no
Athurulu [161]	2017	5% CSPS	5% potassium nitrate toothpaste 3.85% Amine fluoride toothpaste Placebo toothpaste	68 adults -12 weeks	CSPS toothpaste was found to be more effective in reducing DH as the others	yes
Hall [162]	2017	5% CSPS	8% arginine/calcium carbonate toothpaste regular fluoride toothpaste	133 adults -11 weeks	CSPS and arginine toothpastes performed equally in reducing DH	yes
Fu [163]	2019	2.5% CSPS toothpaste	8% arginine toothpaste placebo toothpaste	147 adults -8 weeks	the CSPS and qarginine toothpastes both equally reduced DH more than the control	yes
Alsherbiney [164]	2020	CSPS toothpaste	Zn-carbonate nHAP toothpaste	42 adults -appliances worn with dentin slices	both toothpastes occluded dentin tubules but the HAP toothpaste provided immediate occlusion of dentin tubules	
Bhowmik [165]	2021	NovaMin toothpaste SHY-NM (7.5% CSPS)	Elgydium (fluorinol) toothpaste	30 adults -4 weeks	CSPS toothpaste reduced DH	yes
Ongphichetmetha [166]	2022	5% CSPS	8% arginine	45 adults -2 weeks	SCPS and arginine toothpaste reduce DH	yes
(**d**)						
**Study First Author**	**Year**	**Test** **(CSPS Used)**	**Controls**	**Trial Subjects,** **Duration**	**Outcome**	**Does CSPS Toothpaste Improve Gingival Health?**
Monterubbianesi [79]	2020	CSPS toothpaste	HAP toothpaste herbal toothpaste	25 adults -2 weeks	CSPS toothpaste supported gingival health as well as the herbal toothpaste and better than the HAP toothpaste	yes

**Table 4 biomimetics-08-00331-t004:** (**a**) Tricalcium phosphate (TCP) clinical trials on reducing caries or white spot lesions (WSL) or preventing erosion. (**b**) Tricalcium phosphate (TCP) in situ clinical trials on reducing caries on preventing erosion. (**c**) Tricalcium phosphate (TCP) clinical trials on reducing dentin hypersensitivity. (**d**) Tricalcium phosphate (TCP) clinical trials on improvement of gingival health. (**e**) TCP clinical trials on improving tooth appearance.

(**a**)						
**Study First Author**	**Year**	**Test** **(TCP Used)**	**Controls**	**Trial Subjects,** **Duration**	**Outcome**	**Does TCP Toothpaste Reduce Dental Caries?**
Detsomboonrat [167]	2016	Pureen	1000 ppm fluoride toothpaste 500 ppm fluoride toothpaste	131 mother-child dyads -1 year	caries were reduced by the TCP toothpaste as well as the fluoride toothpastes	yes
(**b**)						
No studies were found.						
(**c**)						
**Study First Author**	**Year**	**Test** **(CSPS Product Used)**	**Controls**	**Trial Subjects,** **Duration**	**Outcome**	**Does TCP Toothpaste** **Desensitize Teeth?**
Jang [168]	2023	Vussen S (190% TCP)	Sensodyne Pleasia (fluoride free)	53 adults -4 weeks	TCP toothpaste effectively reduces DH better than placebo	yes
(**d**)						
No studies were found.						
(**e**)						
No studies were found.						

## Data Availability

The data used in this review was published data in the studies referenced. Online information was referenced and accessed as shown in the reference list. No new data were created.

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
