# Peer review of "Improving Oral Health with Fluoride-Free Calcium-Phosphate-Based Biomimetic Toothpastes: An Update of the Clinical Evidence"

_biomimetics, 2023, doi:10.3390/biomimetics8040331_

Round 1

Reviewer 1 Report

Biomimetics 242826

Improving oral health……….update of the clinical evidence.

H Limeback, J Enas, and F Meyer

This is a timely, well-written manuscript addressing an issue in home health care, that being the health of the oral cavity.  Fluoride proved to be a blessing to reduce carries but has been beleaguered with issues of safety, especially in children where ingestion of home toothpaste is difficult to control. The manuscript avoids the issue of fluoride safety, and rather addresses the need to identify data that can be used for professional recommendations of home care dentifrices that do not contain fluoride. Moreso, the manuscript takes on other pertinent issues by providing a well-constructed approach to reviewing existing for guiding data, allowing oral health care professionals to advise their patients on a case-by-case basis, the only meaningful way to deliver such care. Dr. Elias Zerhouni, former director of the National Institutes of Health, had it right when he advocated for the need for personalized health care.

Some minor issues for the authors to consider are these;

L30, perhaps this construction-"Even in the 21st Century poor oral health remains a major human affliction……."

L43, preventive roles in [home] oral care

L127, could the issues of "qualitative synthesis (risk of bias) and quantitative synthesis (meta-analysis) were not carried out." be considered for a short statement for justification

L128, this reviewer lost track of the pronoun "These" that starts the line.  Could this be clarified with the noun instead?

L135-136 and elsewhere, would the authors and journal consider a table for abbreviations?

L138, The phrase "after rapid screening" seems froth with misconception.  Was it too hard?  Too time consuming?  I doubt these, and the current wording creates an issue that could be avoided.

L148 is an orphan and should be corrected.

L339, what are the "interests"? Financial?, Efficacy?, Risk?, Commercial?.  These were described in the manuscript and may be restated here.

Author Response

Reviewer 1:

L30, perhaps this construction-"Even in the 21st Century poor oral health remains a major human affliction......."

                   This has been changed as suggested.

L43, preventive roles in [home] oral care

                   This has been changed as suggested.

L127, could the issues of "qualitative synthesis (risk of bias) and quantitative synthesis (meta-analysis) were not carried out." be considered for a short statement for justification

                 We have added an explanation why we did not do qualitative and quantitative syntheses of the literature.

L128, this reviewer lost track of the pronoun "These" that starts the line. Could this be clarified with the noun instead?

                 This has been clarified

L135-136 and elsewhere, would the authors and journal consider a table for abbreviations?
                  Please note that wherever abbreviations were used in the text, they were defined in full when they first appeared and then used thereafter. In addition, there was a list of abbreviations in some of the tables. While it might be convenient to have all the abbreviations defined in a separate table, this seems to us to be an extra table that does not need to be added.

L138, The phrase "after rapid screening" seems froth with misconception. Was it too hard? Too time consuming? I doubt these, and the current wording creates an issue that could be avoided.

            This section has been re-written to clarify how we handled the vary large number of Google Scholar ‘results’ when Google Scholar was used as a database.

L148 is an orphan and should be corrected.

               The sentence to which this refers has been clarified and changed.

L339, what are the "interests"? Financial?, Efficacy?, Risk?, Commercial?. These were described in the manuscript and may be restated here.

             The first sentence of the “conclusion” has been rewritten to restate why research on fluoride-free toothpastes has increased and in whose interest.

We would like to thank this reviewer for the generous time in reviewing our manuscript and for such positive feedback.

Reviewer 2 Report

This is a well organized review about non-fluoride biomimetic toothpastes emphasizing their effects on remineralisation,  caries  prevention, improving periodontal health, reducing dentin hypersensitivity, protecting against dental erosion, and safely improving tooth whiting in defining what constitutes improvement in oral health.  Since bioactive materials and biomimetic approach are gaining great interest in the recent years, this review is a good reference for fluoride-free toothpastes used in clinical trials.

The authors included 177 references after searching through PubMed (Medline), Scopus, and Web of Science, as well as Google Scholar, 102 from inception to June 1, 2023 in the review.  

In my humble opinion, this manuscript can be published and will be a good reference for those studying and conducting clinical trials on various toothpastes.

Author Response

Reviewer 2.

There were no recommended changes suggested by this reviewer.

We would like to thank this reviewer for the generous time in reviewing our manuscript and for such positive feedback.

Reviewer 3 Report

Title: Improving oral health with fluoride-free calcium phosphate-based biomimetic toothpastes: an update of the clinical evidence 

This review was conducted to examine the clinical evidence published for fluoride-free calcium phosphate-based toothpastes in order to compare them for determine which one might be a versatile, overall effective toothpaste formulation in promoting good overall oral health.

We thank the authors for their efforts in conducting this study.

Introduction: well written

Methodology: well conducted

I have one minor comment related to the conclusion section.

The conclusion part is too long. It has to be summarized and to the point.

Author Response

Reviewer 3. 

This reviewer suggested we shorten the Conclusion section. This was done.

We would like to thank this reviewers for the generous time in reviewing our manuscript and for such positive feedback.

Round 2

Reviewer 1 Report

The authors revised the majority of issues raised and explained any discrepancies that remain.